# Ultrasonic Features for Evaluation of Adhesive Joints: A Comparative Study of Interface Defects

**DOI:** 10.3390/s24010176

**Published:** 2023-12-28

**Authors:** Damira Smagulova, Bengisu Yilmaz, Elena Jasiuniene

**Affiliations:** 1Ultrasound Research Institute, Kaunas University of Technology, K. Barsausko Str. 59, LT-51423 Kaunas, Lithuaniaelena.jasiuniene@ktu.lt (E.J.); 2Department of Electronics Engineering, Kaunas University of Technology, Studentu Str. 50, LT-51368 Kaunas, Lithuania

**Keywords:** NDT, ultrasonic features, adhesive bonding, interface defects, multiple reflections, detectability

## Abstract

Ultrasonic non-destructive evaluation in pulse-echo mode is used for the inspection of single-lap aluminum adhesive joints, which contain interface defects in bonding area. The aim of the research is to increase the probability of defect detection in addition to ensuring that the defect sizes are accurately estimated. To achieve this, this study explores additional ultrasonic features (not only amplitude) that could provide more accurate information about the quality of the structure and the presence of interface defects. In this work, two types of interface defects, namely inclusions and delaminations, were studied based on the extracted ultrasonic features in order to evaluate the expected feasibility of defect detection and the evaluation of its performance. In addition, an analysis of multiple interface reflections, which have been proved to improve detection in our previous works, was applied along with the extraction of various ultrasonic features, since it can increase the probability of defect detection. The ultrasonic features with the best performance for each defect type were identified and a comparative analysis was carried out, showing that it is more challenging to size inclusion-type defects compared to delaminations. The best performance is observed for the features such as peak-to-peak amplitude, ratio coefficients, absolute energy, absolute time of flight, mean value of the amplitude, standard deviation value, and variation coefficient for both types of defects. The maximum relative error of the defect size compared to the real one for these features is 16.9% for inclusions and 3.6% for delaminations, with minimum errors of 11.4% and 2.2%, respectively. In addition, it was determined that analysis of the data from repetitive reflections from the sample interface, namely, the aluminum-adhesive second and third reflections, that these contribute to an increase in the probability of defect detection.

## 1. Introduction

The development trends and main demands in the automotive and aerospace industries are the reduction in fuel consumption and carbon dioxide emissions as well as the maintenance of high safety for environmental and human health at the same time [1,2,3]. Adhesive bonding is an attractive joining technology that meets the above-listed requirements due to characteristics such as high strength-to-weight ratio, a uniform distribution of loads between joined parts and a decrease in the weight of a structure due to the exclusion of all metal bolts and rivets [2,4,5]. Other advantageous characteristics of adhesively bonded joints are the ability to join similar and dissimilar materials, vibroacoustic damping properties, high fatigue life, impact resistance, residual strength, sealing capabilities and cost effectiveness. However, adhesively bonded joints also have some limitations: inability to disassemble the structure for examination, and the influence of temperature, moisture and other environmental conditions on the bonding strength [2,4,5,6]. During the manufacturing process, the durability of adhesive joints can be affected by poor curing and surface preparation, moisture, contamination, adhesive thickness and overlap length. Therefore, there are two dominant factors: (1) a proper surface treatment ensuring the prevention of contamination; and (2) ensuring the proper curing conditions, which are essential for adhesive bonding strength [1,5,7]. During its service life, the continuous loading of a structure as well as the influence of environmental conditions can cause the occurrence of various defects, including delaminations, disbonds, inclusions, voids, porosity, or weak bonds [4,7,8]. These defects are located at the interface of the structure and are hidden from the human eye. It is not possible to determine their occurrence, presence and assess the severity of these defects on the integrity of the structure via visual inspection. However, the estimation of bonding strength depends on the type of defect present in the adhesive, its extent and location [6]. To detect defects in such multi-layered adhesively bonded structures is a challenging task. Despite all the advantages that adhesive bondings have, the application of this technology in the aerospace industry is still limited due to a lack of knowledge on the reliability of the non-destructive testing (NDT) techniques [2,7].

Different methods of NDT are used to test adhesively bonded joints. For example, in order to detect defects such as porosity and voids, X-ray radiography is well suited. Some recent works have been performed to characterize such defects in adhesively bonded joints [3,5,9,10,11]. Unfortunately, this method is too hazardous to human health due to radiation, and it is financially costly, especially for industry use. For non-volumetric defects such as delaminations and disbonds, a more common NDT method with high performance is a conventional ultrasound. Various configurations of ultrasonic methods based on the measurement of the difference in amplitude of reflected signals from defective and non-defective interfaces have been used to detect interface defects [12,13,14,15,16]. In addition, the following ultrasonic features, which characterize the signal response, were extracted to evaluate the adhesive quality: phase shift [12], ratio coefficients of peak-to-peak amplitudes [2], the decibel drop between ultrasonic pulses [17,18], maximum correlation between pulse response and excitation signals to obtain time reversal signals of Lamb waves [19], mean value, root mean square, standard deviation, skewness, kurtosis and crest factor, as well as modal frequency, modal mode and damping [20]. A number of works have been carried out using numerical evaluations and modeling to study the feasibility of different NDT methods [2,7,21,22]. Advanced post-processing algorithms of the collected data have been developed to provide C-scan images of the interface defects in multilayer structures, improve detection quality or extract features from the signal responses [2,3,16,19,20,23]. Furthermore, machine learning is also being applied to study the defects in the structures [1,24,25,26]. In our previous works [2], a novel post-processing technique was developed for the detection of disbonds in multilayered structures and it eliminated some influential factors and improved detectability. The work of Jakub Kowlaczyk et al. [18] also investigated influential factors such as longitudinal wave velocity, elastic properties, the surface roughness, anisotropy and microstructure of the material as well as attenuation. Jasiuniene et al. [3] proposed a multidimensional fusion technique that combined the resultant data received from ultrasonic and X-ray inspections for adhesively bonded components used in aerospace. Due to the fusion of the data from different NDT methods, the probability of bonding defect detection was improved. Samaitis et al. [1] applied linear ultrasound and machine-learning algorithms to detect weak bonds in adhesive joints. In this work, it was shown that a classical ultrasonic pulse-echo method along with LDA (linear discriminant analysis) feature transformation and SVM (support vector machine) classifier algorithms can detect deviations in bonding integrity. LDA explores the importance of extracted features and classification accuracy, and then SVM compares the accuracies between the different features. Yilmaz et al. [15] carried out a comparison of the performance of different ultrasonic NDT methods in the evaluation of bonding quality in multilayered structures. The following methods, including bulk waves and guided waves, were studied: contact, immersion and air-coupled. As a result, some methods, such as air-coupled guided waves, performed well in detecting the presence of the defects, whereas bulk waves showed higher performance in defect sizing. Each method has its own advantages and disadvantages; therefore, the selection of the method depends on the objective to be achieved. Wojtczak et al. [27] investigated CFRP/steel adhesive joints using ultrasonic guided waves. Three-staged algorithms were proposed for damage imaging: estimation of dispersion curves of the component, novel weighted root mean square (WRMS) parameters for damage identification and the recording of guided wavefields by laser vibrometer. Recently, Rao et al. [28] proposed an ultrasonic method based on the supervised fully convolution method (FCN) to reconstruct quantitatively hidden disbonds in multilayered structures with high contrast. This method reconstructs longitudinal wave velocity models from the measured ultrasonic data by extracting features from the component. In addition, many other works have been performed using NDT methods, such as ultrasonic guided waves [4,12,20,29,30,31,32], thermography [10,33], non-linear ultrasound [34], laser ultrasound [8,31,35] and eddy current [36], to investigate the integrity of adhesive bonding.

However, the reliability of the inspection techniques still remains challenging due to many aspects of the specific sample, the defect type under investigation, as well as many other influential factors such as the geometry of the sample, multilayered structure, thickness of adherends and adhesive layer, different elastic and acoustic properties, surface and internal layer roughness and unevenness, defect structure and position in the adhesive layer, as well as ultrasonic wave interference, reverberations and overlapping. The development of a technique that will take into account all these factors and will produce high efficiency for the detection of interface defects, where it is also possible to apply various algorithms for data processing, with the possible use of artificial intelligence, and the automation and optimization of processes represent an up-to-date task [2,3,15].

In this work, the detection and accurate sizing of interface defects in adhesively bonded aluminum joints such as inclusions and delaminations is of interest. This study was motivated by the lack of a reliable method of testing adhesive joints that could guarantee high reliability and performance, and which could increase the usage of bonded structures in the aeronautical industry. Therefore, the scientific objective of the work is to increase the probability of detection of interface defects in adhesive joints of multilayer structures. Despite the progress made in the ultrasonic evaluation of adhesive joints, there are still factors that affect the detectability of defects in multilayer structures that have not yet been eliminated, leading to the need to improve the assessment of bond integrity. In order to fill this gap and contribute to the detectability of inclusion and disbond types of defects, signal responses utilizing signal processing for various ultrasonic feature extraction were studied. In order to achieve the main objective, the features were determined based on the knowledge of wave propagation physics and expertise; the performance of each was evaluated and a comparative analysis was carried out. Thus, ultrasonic non-destructive inspection of aluminum single-lap joints with different types of bonding defects was carried out. Then, ultrasonic (UT) features which are capable of increasing the probability of defect detection were determined and extracted. Then, an evaluation and analysis of the performance of the extracted ultrasonic features, with respect to different types of bonding defects, were performed.

## 2. Materials and Methods

This section provides an overview of the characteristics of joints of adhesively bonded aluminum plates and outlines the workflow employed for the investigation. Signal propagation through the layers of the structures was modeled to discern the interaction of ultrasonic waves with various materials, interfaces and defects in the structures. Time instants for signal reflections from the interfaces of the samples were calculated. Subsequently, an ultrasonic experimental investigation was conducted to collect the data for further post-processing. The methodology for analyzing multiple interface reflections with assessed time intervals is also presented. Additionally, the determination and extraction of ultrasonic features, which can improve the probability of interface defect detection, are also described in this section.

### 2.1. Sample Description

Aluminum single-lap joints manufactured at Cotesa GmBH, Germany, were used for the investigation. Two plates of aluminum 2024 were bonded using a 3M Scotch-Weld structural adhesive film AF 163 k. One sample was produced with brass inclusions between the aluminum sheet and the adhesive layer and another sample was produced with delaminations on the interface. The delaminations were produced using the Wrigtlon 4600 release film. The release film was double-folded in order to obtain an air gap between the two layers. The defects were rectangular. Defects in the case of brass inclusions and delaminations were the same size, 12.7 mm in length and width. The dimensions of the sample and the defects are presented in Table 1. Pictures of both samples are shown in Figure 1 and the schematics of the joint are shown in Figure 2.

### 2.2. Signal Modeling

Ultrasonic signal modeling was performed in order to collect the data, analyze the propagation of ultrasonic waves in the structures, learn about the interaction of ultrasonic waves with defects vs. perfect bonding, and determine what techniques of post-processing could be applied to increase the probability of defect detection. The propagation of the signal through the sample of the aluminum lap joint with interface defects was modeled using MATLAB R2023a software. The paths of ultrasonic wave propagation and reflection from each structure layer with corresponding time moments, as well as pulse responses, are shown in Figure 3. The time moments of the arrival time of reflections are presented in Table 2.

Due to theoretical calculations and the modeling of ultrasonic wave propagation paths for both samples with different defect types, it was possible to reconstruct the sequence and order of the expected reflections from the boundaries. The obtained pulse response plots characterized the arrival time of the reflected signals. The positive and negative polarity of the impulses characterizes the difference in the acoustic characteristics of the material of each sample layer and phase change [2,23]. In the case of the joint with brass inclusions (Figure 3a), it can be observed that the time difference between reflections is quite small, which explains the occurrence of the signal overlap. This fact can make it much more difficult to detect defects. In the case of the joint with delaminations, it can be observed that ultrasonic waves reflect fully from the aluminum–delamination interface. This can be explained by the fact that defects such as delaminations or disbonds are characterized by the total debonding of two glued layers, implying the presence of air. As a result, the time difference between the multiple reflections from aluminum–delamination interface is 0.49µs, which is still a very small value. Therefore, it is very important to take into account the thickness of the upper layers before the defect and select the appropriate frequency of the transducer as well as to have knowledge of the sequence, order and time of arrival of the reflected signals for further processing [2,23].

### 2.3. Ultrasonic Non-Destructive Evaluation

Aluminum single-lap joints with brass inclusions and delaminations were inspected using a pulse-echo ultrasonic technique. The inspection in immersion mode was conducted at the Ultrasound Research institute. The TecScan measurement system was used to carry out experiments. The sample was fixed on a special table and immersed in a tank of water. A 15 MHz focused transducer was placed in front of the sample and fixed perpendicular to the sample surface. Additionally, the transducer perpendicularity and distance to the surface zone were adjusted by observing the maximum amplitude of the reflected signal. This configuration was used for scanning and recording the data, since it provides focus on the sample surface and is able to provide results with higher detectability. The scanning area was set in 2 directions with a step of 0.2 mm. The inspection set-up is shown in Figure 4.

After conducting the experimental investigation, the data were collected and analyzed using MATLAB software. The A-scans at the selected points of the sample surface were analyzed and displayed. From the previous works performed [2], it was found that an analysis of multiple interface reflections one by one improves the probability of interface defect detection. Therefore, in this study, the performance of each ultrasonic feature will be investigated for the time intervals that correspond to multiple interface reflections, which were determined using signal modeling (Figure 3; Table 2). A-scans with indicated time intervals for multiple interface reflections are shown in Figure 5.

### 2.4. Determination and Extraction of Ultrasonic Features

In this section, different ultrasonic features, which can have an influence/impact on defect detection, were determined based on wave propagation physics knowledge and expertise, in order to evaluate their performance and to identify features and their subsets with the best reliability [1,3]. Different features were extracted from time and frequency domains and then analyzed.

In total, the following ultrasonic features (Table 3) were determined and extracted from the measurement data:Peak-to-peak amplitude;Ratio coefficients;Attenuation;Maximum amplitude in the frequency domain;Absolute energy;Frequency value at the maximum amplitude;Absolute time of flight difference;Kurtosis (tailedness);Mean value of the amplitude in the frequency domain;Skewness;Standard deviation value in the time domain;Standard deviation value in the frequency domain;Variation coefficient in the time domain;Variation coefficient in the frequency domain.

Ultrasonic features and their mathematical expressions are presented in Table 3. All listed ultrasonic features were calculated in time intervals of 4 repeated multiple reflections from the interface (Figure 5). An example of the calculation of the peak-to peak amplitude feature in time intervals of multiple interface reflections is shown in Figure 6. The remaining ultrasonic features were calculated accordingly. The ratio coefficients were calculated as the ratio of amplitudes of different time intervals of reflected signals from the aluminum–defect interface. The kurtosis feature characterizes whether the data are heavy-tailed or light-tailed in comparison to the normal distribution. Skewness measures the lack of symmetry of the signal. The standard deviation value measures how the data of amplitude in relation to mean value are dispersed. The variation coefficient value is a value between the standard error and the mean value. In the case of the time of flight feature, different thresholds were applied: 2%, −2%, 10% and −10%. The different values of thresholds were selected to calculate the time of flight within the gate at different levels of signal crossing.

## 3. Results and Discussion

All the listed features were extracted in the case of the samples with brass inclusions and delaminations. The evaluation of each ultrasonic feature performance was performed using the −6 dB drop method. This method includes monitoring the reflected signal from the defective interface, which first reaches its maximum and then drops to 50% at the edges of the signal height. The point where the signal is located at half of the maximum peak of the ultrasonic A-scan is used to estimate the length of the defect. Consequently, flaw lengths were sized in the middle of the defects along the x axis. C-scans and slices of the intensity variation in the middle of the defects along the x axis (red dashed line), used to size each defect, are presented in Figure 7. In addition, the −6 dB level on the intensity variation in Figure 7 is also indicated with a red dashed line. These figures provide examples of defect sizing performed for peak-to-peak and variation coefficient features. The resulting C-scans of ultrasonic features, which presented the highest performance for both samples, as well as estimated the mean value of relative error of defect sizing compared to reference defect size, are presented in Table 4 and Table 5, respectively. During the experimental investigation, the measurement was performed 128 times and averaged in the TecScan system. The mean value of the relative error and the error range (standard error of the mean) were calculated on the basis of five measurements of the defects. The relative error percentage is calculated according to the equation
(1)$$\%error=\frac{\left|V_{measured}-V_{real}\right|}{V_{real}}\times 100\%$$
where $V_{measured}$ is the experimentally received value and $V_{real}$ is the reference size of the defect.

Observing the C-scans reveals that some defects have a more complex structure. For example, brass inclusion No. 4 consists of double layers of brass, where the second layer has a rectangular shape of a smaller size. In the case of delamination No. 1, the amplitude variation in ultrasonic reflection drops in the center of the defect. The reason for such an effect may be the defect structure and/or the entire sample structure above this defect. Thus, such factors can affect the detection of defects as well as the ability to measure the defect dimensions. In this case, another technique to measure the sizes of the defects might be required, or some post-processing algorithms which will be able to eliminate the influence of these uncertain parameters can be developed [2]. However, the primary focus of this work was to investigate the performance of each ultrasonic feature. Therefore, defects in both samples were analyzed, compared and the results are presented in this paper.

From the calculated mean relative errors in the case of the sample with brass inclusions, the following ultrasonic features showed the best performance: absolute time of flight ∆t between *t_i_* − 2 and *t_i_* − 3 at the 2% threshold, variation coefficient in frequency domain ${cv}_{f}$ at fourth multiple interface reflection t_i_ − 4, kurtosis k at *t_i_* − 1, ratio coefficients $K_{2}$ of *t_i_* − 3 and *t_i_* − 2 interface reflection and $K_{2}$ of *t_i_* − 3 and *t_i_* − 1 time intervals. In the case of the sample with delaminations, these features performed the best: absolute time of flight ∆t between *t_i_* − 2 and *t_i_* − 3 at the 2% threshold, peak-to-peak amplitude $U_{pp}$ at *t_i_* − 3, mean value of the amplitude in frequency domain $u_{fmean}$ at *t_i_* − 3, variation coefficient in time domain cv at *t_i_* − 3 and standard deviation value in time domain σ at *t_i_* − 3 time interval. As a result, the absolute time of flight ∆t between *t_i_* − 2 and *t_i_* − 3 at the 2% threshold demonstrated the highest performance for both types of the defect.

Bar graphs of all relative errors estimated for each ultrasonic feature are presented in Figure 8 and Figure 9. The results of defect size error calculation at multiple interface reflections are presented in Table 6 and compared.

It was determined that the performance of the ultrasonic features also depends on the number of repetitive interface reflections from where these features are calculated. Therefore, the tables above present appropriate numbers of interface reflections along with the features. Eventually, in this work, it was demonstrated that an analysis of multiple reflections from the interface provides a significant contribution in improving defect detectability [2,3,23]. Upon comparing the bar graphs of calculated relative errors for aluminum joints with brass inclusions, it was observed that, mostly by studying second multiple interface reflection *t_i_* − 2, the sizing of the brass inclusions is better compared to other time intervals. The third and fourth interface reflections *t_i_* − 3 and *t_i_ −* 4 have a higher relative error. This can be caused by the attenuation of the signal within multi-reflections from the interface. In the case of the joint with delaminations, a better performance is observed for second and third multiple interface reflections depending on the feature. The exceptions are the variation coefficient in frequency domain ${cv}_{f}$ and skewness s, which showed the best performance at *t_i_* − 4, kurtosis k and absolute energy A at *t_i_* − 1 for brass inclusions. The variation coefficient in frequency domain ${cv}_{f}$ and kurtosis k at *t_i_* − 4 as well as skewness s (*t_i_* − 1) have also demonstrated high performance for sizing of delaminations.

Based on the obtained results, certain ultrasonic features demonstrate high performance for the sizing of both inclusions and delaminations. These features are peak-to-peak amplitude $U_{pp}$, ratio coefficients $K_{2}$, absolute time of flight ∆t, absolute energy A, mean value of the amplitude in frequency domain $u_{fmean}$, standard deviation value in time and frequency domains σ and $\sigma_{f}$, and variation coefficient in time and frequency domain cv and ${cv}_{f}$. In the case of absolute time of flight features, different thresholds were applied: 2%, −2%, 10% and −10%. Quite promising results were obtained for the absolute time of flight feature ∆t between *t_i_* − 2 and *t_i_* − 3 at the 2% threshold for brass inclusions and delaminations. Ratio coefficients of the peak-to-peak amplitudes of repetitive muti-reflections demonstrated consistent high efficiency for different types of defects. In the previous work [2], these coefficients also helped to improve the probability of disbond detection located in the middle of the adhesive layer in dissimilar material joints. In addition, $U_{pp}$, $u_{fmean}$, σ and $\sigma_{f}$ as well as cv show reliable results for almost all multiple reflections that were considered. This could mean that these parameters will consistently exhibit high performance across all types of defects [1,3,20]. However, some ultrasonic features have high performance for the detection of brass inclusions and low performance for the detection of delaminations and vice versa.

From the results obtained, it was revealed that the higher the mean relative error of the features, the greater the dispersion of standard error of the mean in cases of both types of defects. The dispersion of standard error of the mean is affected by the influential factors that appear in the structure during inspection, such as ultrasonic wave interference in the inner layers, overlapping and signal reverberation. In addition, the structure of the sample and the defects, as well as their materials, also influence the accuracy of the results.

Moreover, sizing is more accurate in the case of the investigation sample with delaminations. The lowest relative error is 2.2%, while for the brass inclusion sample, the lowest relative error is 11.4%. Such a significant difference can be explained by the material characteristics of the defect. Since delaminations are characterized by the presence of air, the acoustic impedance mismatch of structure materials is higher and more ultrasonic energy reflects on such interfaces, returning back to the receiving sensor. Brass material does not have a high difference in impedance compared to aluminum; therefore, more ultrasonic energy is being transmitted through the aluminum–brass interface and less travels back to the transducer [2,13].

This comparison of the results obtained with those of previous studies underlines the consistency and reliability of the identified ultrasonic features for defect evaluation in multilayered structures.

## 4. Conclusions

The purpose of this work was to select ultrasonic features that would enable us to size different bonding defects in adhesively bonded aluminum joints more accurately. For this, different valuable ultrasonic features which can have the ability to increase probability of detection were determined, extracted and their performance was analyzed. In this study, emphasis was placed on determining ultrasonic features with high performance for inclusions and delaminations that are typical for multilayered structures.

The following was determined from the presented results:
It is more difficult to correctly size inclusion-type defects in the adhesive layer compared to delaminations due to the similar acoustic properties of the defects;The lowest error for delamination detection is 2.2%, and for brass inclusions it is 11.4%. The maximum errors of the features that performed the best are 3.6% and 16.9%, respectively;Ultrasonic features that showed high performance for both types of defects are as follows: peak-to-peak amplitude $U_{pp}$, absolute time of flight ∆t, ratio coefficients $K_{2}$, absolute energy A, mean value of the amplitude in frequency domain $u_{fmean}$, standard deviation value in time and frequency domains σ and $\sigma_{f}$, and variation coefficients in the time and frequency domain cv and ${cv}_{f}$;In the case of brass inclusions, kurtosis k at *t_i_* − 1 and maximum amplitude at frequency domain $U_{fmax}$ *t_i_* − 4 also showed quite high performance, while in the case of delaminations, the variation coefficient in time domain cv at *t_i_* − 2 and at *t_i_* − 3 time intervals showed high performance.The exploration of first interface reflection has the lowest possibility of correctly sizing the defect. However, the defect presence is identified. For the sizing, second and third interface reflections show better performance in the case of inclusions and delaminations, respectively. The fourth reflection is characterized by signal damping and a decrease in the performance of ultrasonic features.

The results obtained can be used for the optimization of the technique for the detection and sizing of interface defects as well as for the development of automated algorithms. In further research, we plan to investigate weak bonds in multilayered structures in the same way and compare the results with those obtained in this work.

## Figures and Tables

**Figure 1 sensors-24-00176-f001:**
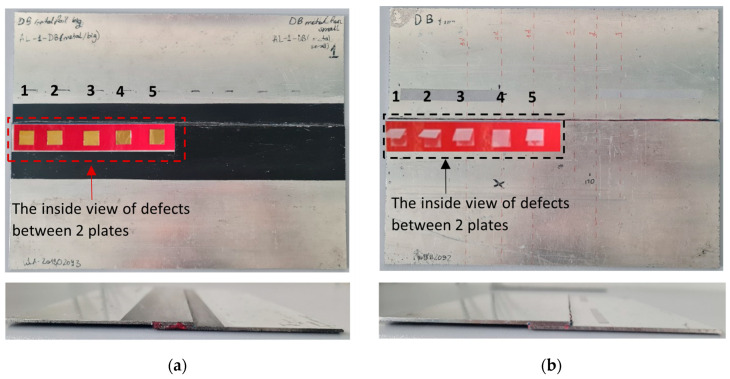
Pictures of single-lap joints with (**a**) brass inclusions; and (**b**) delaminations.

**Figure 2 sensors-24-00176-f002:**
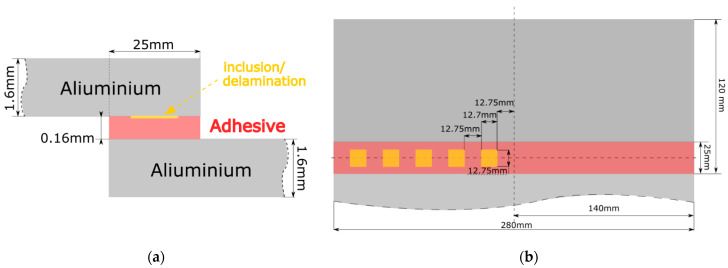
Schematics of single-lap joints with brass inclusions and delaminations: (**a**) side view; and (**b**) top view.

**Figure 3 sensors-24-00176-f003:**
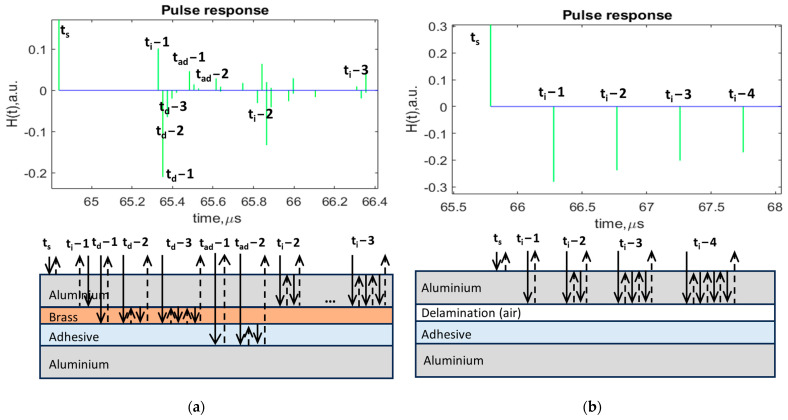
Pulse responses of the signal reflections from the sample boundaries and ultrasonic wave propagation paths: (**a**) aluminum joint with brass inclusions; and (**b**) aluminum joint with delaminations.

**Figure 4 sensors-24-00176-f004:**
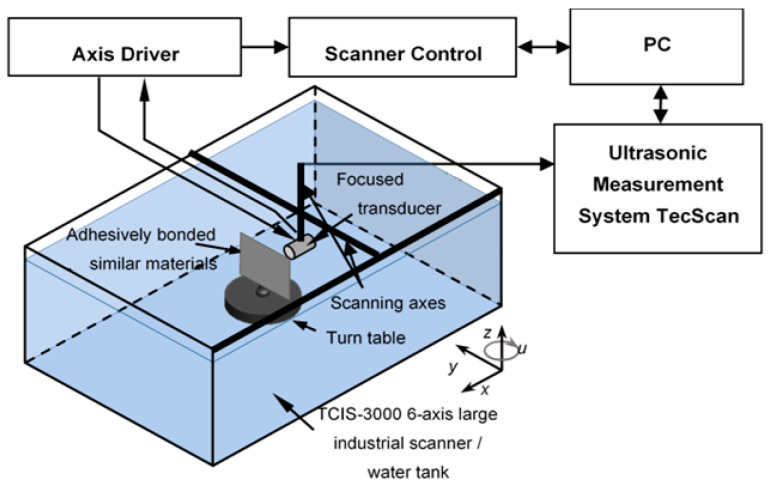
Set-up of ultrasonic inspection of aluminum lap joint [2].

**Figure 5 sensors-24-00176-f005:**
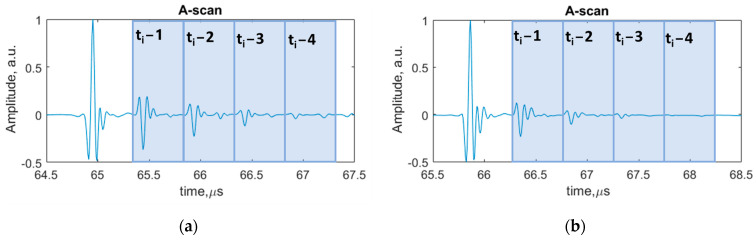
A-scans with indicated time intervals of each multiple reflections: (**a**) sample with brass inclusions; (**b**) sample with delaminations.

**Figure 6 sensors-24-00176-f006:**
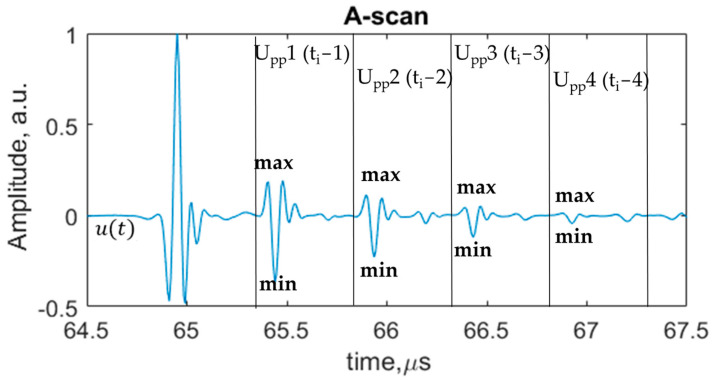
Calculation of peak-to-peak amplitude feature at time intervals of multiple interface reflections.

**Figure 7 sensors-24-00176-f007:**
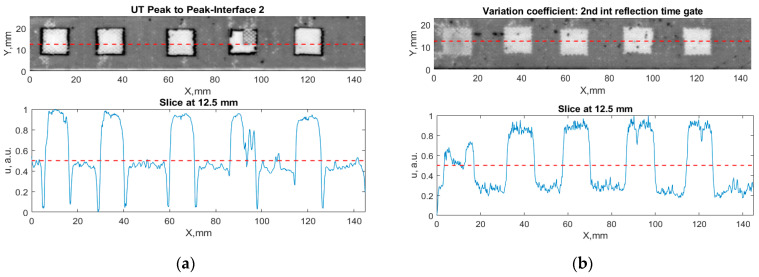
C-scans and slices of intensity variations along the x axis of the −6 dB drop method for the (**a**) sample with brass inclusions; and (**b**) sample with delaminations.

**Figure 8 sensors-24-00176-f008:**
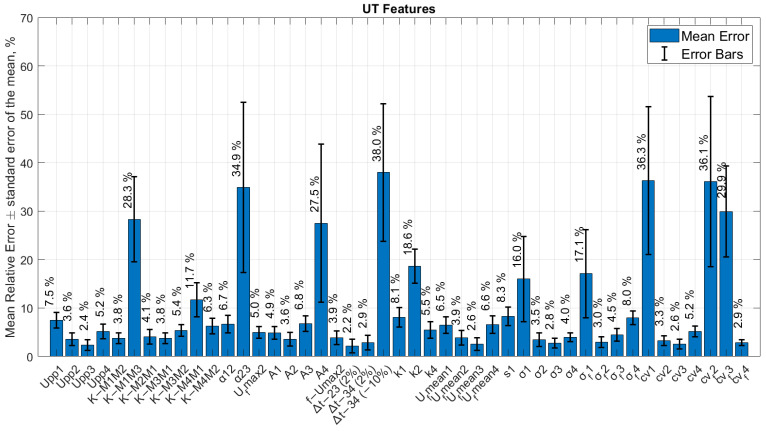
The bar graph of all ultrasonic features and relative errors of defect size (No. 3) in the sample with delaminations.

**Figure 9 sensors-24-00176-f009:**
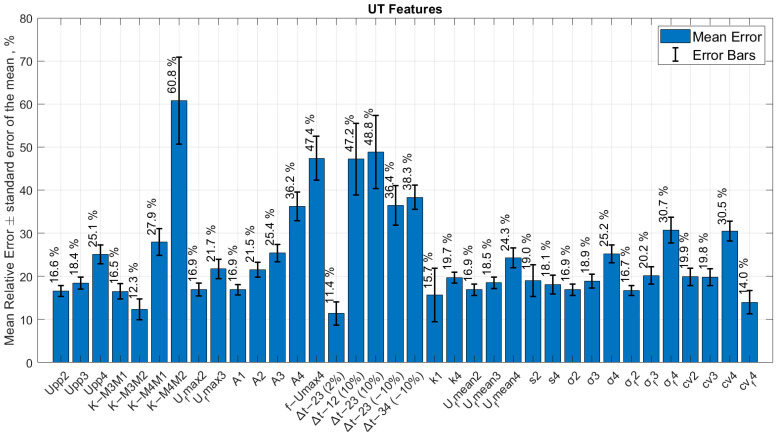
The bar graph of all ultrasonic features and relative errors of defect size (No. 3) in the sample with brass inclusions.

**Table 1 sensors-24-00176-t001:** Sample and defect dimensions.

Material	Length, mm	Width, mm	Thickness, mm
Aluminum joint	280	215	3.36
Aluminum plate	280	120	1.6
Adhesive film	280	25	0.16
Defects	12.7	12.7	N/A

**Table 2 sensors-24-00176-t002:** The time moments of the arrival time of reflections s.

Name of Signal Reflection	Time Moment for the Sample with Inclusions, µs	Time Moment for the Sample with Delaminations, µs
*t_s_*	64.84	65.79
*t_i_* – 1	65.33	66.28
*t_d_* – 1	65.35	-
*t_d_* – 2	65.37	-
*t_d_* – 3	65.39	-
*t_ad_* – 1	65.48	-
*t_ad_* – 2	65.62	-
*t_i_* – 2	65.82	66.77
*t_i_* – 3	66.31	67.26
*t_i_* – 4	66.80	67.75

**Table 3 sensors-24-00176-t003:** Mathematical expressions of ultrasonic features.

No	Ultrasonic Feature	Mathematical Expression
1	Peak-to-peak amplitude, $U_{pp}$	$U_{pp}=\max\left(u\left(t\right)\right)-min(u\left(t\right))$, $t\in t_{n}÷t_{n+1}$, n = 1,2,3,4 (interface reflections)
2	Ratio coefficients, $K_{1},\;K_{2}$	$K_{1}=\frac{U_{{pp}_{n}}}{U_{{pp}_{n+1}}},\;K_{2}=\frac{U_{{pp}_{n+1}}}{U_{{pp}_{n}}}$
3	Attenuation, α	$\alpha =20{log}_{10}\frac{U_{{pp}_{n}}}{U_{{pp}_{n+1}}}$
4	Maximum amplitude at frequency domain, $U_{fmax}$	$U_{fmax}=max(FT\left(u\left(t\right)\right)),$ $t\in t_{n}÷t_{n+1}$, FT—Fourier Transform
5	Absolute Energy, A	$A=\sum_{t_{n}}^{t_{n+1}}{U_{p-p}}^{2}$
6	Frequency value at the maximum amplitude, $f_{Umax}$	$f_{Umax}=FT\left(u\left(t\right)\right),$ $t\in t_{n}÷t_{n+1}$
7	Absolute time of flight difference, ∆t	$∆t=\left|t_{n+1}-t_{n}\right|$
8	Kurtosis, k	$k=FT\frac{E{\left(u(t_{n}÷t_{n+1}\right)-\mu )}^{4}}{\sigma^{4}},$ μ—is a mean of $\left(u(t_{n}÷t_{n+1})\right)$, σ is a standard deviation, E is the expected value of the quantity ${\left(u(t_{n}÷t_{n+1}\right)-\mu )}^{4}$
9	Mean value of the amplitude in frequency domain, $u_{fmean}$	$u_{fmean}=\sum_{i=1}^{N}FT\left(\frac{u_{i}(t)}{N}\right),$ $t\in t_{n}÷t_{n+1}$, $u_{i}$—is each datum of amplitudes at selected time interval, N—is a number of observations
10	Skewness, s	$s=FT\frac{E{\left(u(t_{n}÷t_{n+1}\right)-\mu )}^{3}}{\sigma^{3}}$ μ—is a mean of $\left(u(t_{n}÷t_{n+1})\right)$, σ is a standard deviation, E is the expected value of the quantity${\left(u(t_{n}÷t_{n+1}\right)-\mu )}^{3}$
11	Standard deviation value in time domain, σ	$\sigma =\sqrt{\frac{1}{N-1}\sum_{i=1}^{N}u_{i}(t)-\bar{u(t)}}$ $u_{i}$—is each data of amplitudes at selected time interval, $\bar{u(t)}$*—*is a mean value, *N*—is a number of observations
12	Standard deviation value in frequency domain, $\sigma_{f}$	$\sigma_{f}=FT\cdot \sqrt{\frac{1}{N-1}\sum_{i=1}^{N}u_{i}(t)-\bar{u(t)}}$
13	Variation coefficient in time domain, cv	$cv=\frac{\sigma }{u_{mean}}$
14	Variation coefficient in frequency domain, ${cv}_{f}$	${cv}_{f}=FT\left(\frac{\sigma_{f}}{u_{fmean}}\right)$

**Table 4 sensors-24-00176-t004:** Performance of ultrasonic features for the aluminum joint with delaminations and corresponding C-scans.

Ultrasonic Features	No Interface Reflection	Relative Error, %	C-Scans of Extracted Ultrasonic Features
∆t	*t_i_* − 2 and *t_i_* − 3 (2%)	2.2	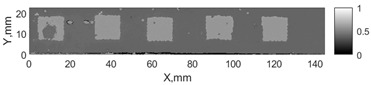
$U_{pp}$	*t_i_* − 3	2.4	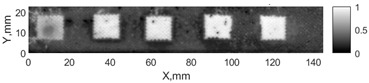
$u_{fmean}$	*t_i_* − 3	2.6	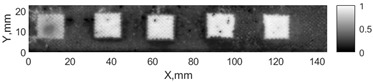
cv	*t_i_* − 3	2.6	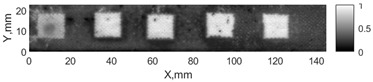
σ	*t_i_* − 3	2.8	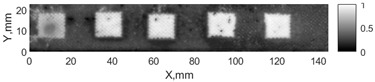
∆t	*t_i_* − 3 and *t_i_* − 4 (2%)	2.9	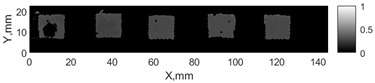
${cv}_{f}$	*t_i_* − 4	2.9	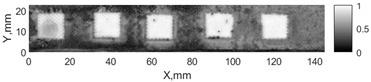
$\sigma_{f}$	*t_i_* − 2	3.0	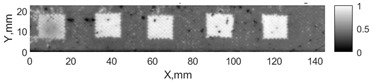
cv	*t_i_* − 2	3.3	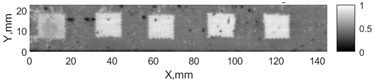
σ	*t_i_* − 2	3.5	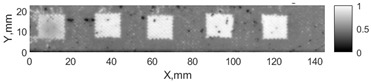
$U_{pp}$	*t_i_* − 2	3.6	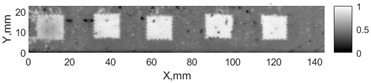
A	*t_i_* − 2	3.6	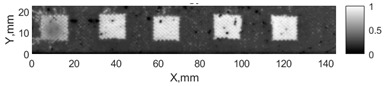

**Table 5 sensors-24-00176-t005:** Performance of ultrasonic features for the aluminum joint with brass inclusions and corresponding C-scans.

Ultrasonic Features	No interface Reflection	Relative Error, %	C-Scans of Extracted Ultrasonic Features
∆t	*t_i_* − 2 and *t_i_* − 3 (2%)	11.4	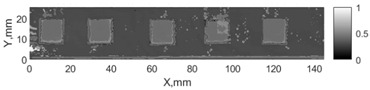
$K_{2}$	*t_i_* − 3 and *t_i_* − 2	12.3	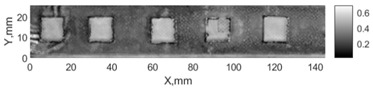
${cv}_{f}$	*t_i_* − 4	14.0	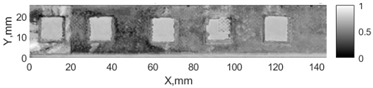
k	*t_i_* − 1	15.7	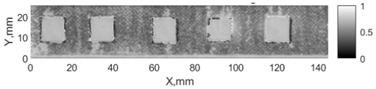
$K_{2}$	*t_i_* − 3 and *t_i_* − 1	16.5	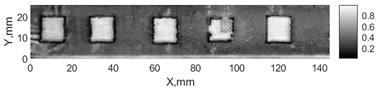
$U_{pp}$	*t_i_* − 2	16.6	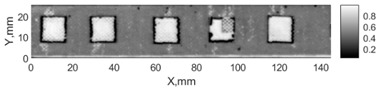
$\sigma_{f}$	*t_i_* − 2	16.7	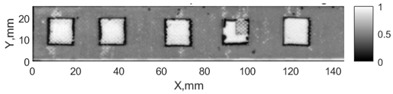
σ	*t_i_* − 2	16.9	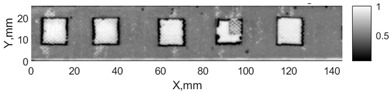
$U_{fmax}$	*t_i_* − 2	16.9	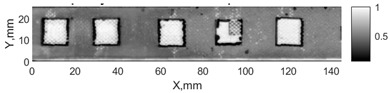
$u_{fmean}$	*t_i_* − 2	16.9	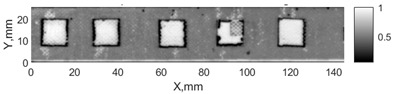
A	*t_i_* − 1	16.9	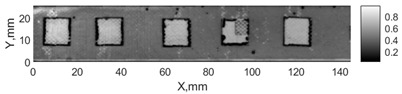
s	*t_i_* − 4	18.1	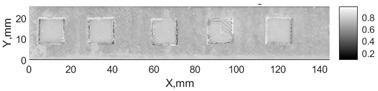

**Table 6 sensors-24-00176-t006:** Relative sizing error of the defects in aluminum joints with brass inclusions and delaminations.

Ultrasonic Feature	No. of Interface Reflection	Brass Inclusions Sample		Delaminations Sample	
		Mean Error,%	Error Range,%	Mean Error,%	Error Range,%
$U_{pp}$	*t_i_* − 1	-	-	7.5	±1.6
	*t_i_* − 2	16.6	±1.3	3.6	±1.3
	*t_i_* − 3	18.4	±1.4	2.4	±1.1
	*t_i_* − 4	25.1	±2.2	5.2	±1.5
$K_{1}$	*t_i_* − 1 and *t_i_* − 2	-	-	3.8	±1.1
	*t_i_* − 1 and *t_i_* − 3	-	-	28.3	±8.8
$K_{2}$	*t_i_* − 2 and *t_i_* − 1	-	-	4.1	±1.5
	*t_i_* − 3 and *t_i_* − 1	16.5	±1.8	3.8	±1.1
	*t_i_* − 3 and *t_i_* − 2	12.3	±2.4	5.4	±1.2
	*t_i_* − 4 and *t_i_* − 1	27.9	±3.1	11.7	±3.5
	*t_i_* − 4 and *t_i_* − 2	60.8	±10.1	6.3	±1.6
α	*t_i_* − 1 and *t_i_* − 2	-	-	6.7	±1.8
	*t_i_* − 2 and *t_i_* − 3	-	-	34.9	±17.6
$U_{fmax}$	*t_i_* − 2	16.9	±1.5	5.0	±1.2
	*t_i_* − 3	21.7	±2.2	-	-
A	*t_i_* − 1	16.9	±1.2	4.9	±1.3
	*t_i_* − 2	21.5	±1.7	3.6	±1.4
	*t_i_* − 3	25.4	±2.0	6.8	±1.6
	*t_i_* − 4	36.2	±3.3	27.5	±16.3
$f_{Umax}$	*t_i_* − 2	-	-	3.9	±1.4
	*t_i_* − 4	47.4	±5.1	-	-
∆t	*t_i_* − 2 and *t_i_* − 3 (2%)	11.4	±2.7	2.2	±1.4
	*t_i_* − 3 and *t_i_* − 4 (2%)	-	-	2.9	±1.5
	*t_i_* − 1 and *t_i_* − 2 (10%)	47.2	±8.3	-	-
	*t_i_* − 2 and *t_i_* − 3 (10%)	48.8	±8.5	-	-
	*t_i_* − 2 and *t_i_* − 3 (−10%)	36.4	±4.6	-	-
	*t_i_* − 3 and *t_i_* − 4 (−10%)	38.3	±2.8	38	±14.2
k	*t_i_* − 1	15.7	±6.2	8.1	±2.0
	*t_i_* − 2	-	-	18.6	±3.5
	*t_i_* − 4	19.7	±1.3	5.5	±1.7
$u_{fmean}$	*t_i_* − 1	-	-	6.5	±1.7
	*t_i_* − 2	16.9	±1.3	3.9	±1.5
	*t_i_* − 3	18.5	±1.3	2.6	±1.3
	*t_i_* − 4	24.3	±2.3	6.6	±1.8
s	*t_i_* − 1	-	-	8.3	±1.9
	*t_i_* − 2	19.0	±3.7	-	-
	*t_i_* − 4	18.1	±2.2	-	-
σ	*t_i_* − 1	-	-	16	±8.8
	*t_i_* − 2	16.9	±1.3	3.5	±1.4
	*t_i_* − 3	18.9	±1.6	2.8	±1.0
	*t_i_* − 4	25.2	±2.1	4.0	±0.9
$\sigma_{f}$	*t_i_* − 1	-	-	17.1	±9.1
	*t_i_* − 2	16.7	±1.2	3.0	±1.1
	*t_i_* − 3	20.2	±2.0	4.5	±1.3
	*t_i_* − 4	30.7	±3.0	8.0	±1.4
cv	*t_i_* − 1	-	-	36.3	±15.3
	*t_i_* − 2	19.9	±2.0	3.3	±1.0
	*t_i_* − 3	19.8	±1.9	2.6	±1.0
	*t_i_* − 4	30.5	±2.3	5.2	±1.1
${cv}_{f}$	*t_i_* − 2	-	-	36.1	±17.6
	*t_i_* − 3	-	-	29.9	±9.4
	*t_i_* − 4	14.0	±2.7	2.9	±0.6

## Data Availability

The data presented in this study are available on request from the corresponding author.

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
