# Peer review of "Ultrasonic Features for Evaluation of Adhesive Joints: A Comparative Study of Interface Defects"

_sensors, 2023, doi:10.3390/s24010176_

Round 1

Reviewer 1 Report

Comments and Suggestions for Authors

Dear Authors,

the manuscript entitled "Ultrasonic Features for Evaluation of Adhesive Joints: A Comparative Study of Interface Defects" by Damira Smagulova and co-authors deals with the investigation of adhesive joint (aluminium-adhesive with flaw-aluminium). In my opinion, the research results presented in the article have significant practical applications in assessing the quality of multi-layer adhesive joints currently used in the automotive industry. The Authors performed simulation tests of ultrasonic wave propagation with a frequency of 15 MHz, as well as tests on a sample with artificially made defects in the adhesive joint.  I appreciate the contribution that the Authors made in simulation, experimental testing as well as preparing the manuscript. However, in my opinion the manuscript needs to be significantly improved in some fields and some general remarks as well as the specific comments are bellow.

Evaluation of the paper, general remarks, editorial comments/typos:

1) Abstract section should present quantitative results and not only the most important qualitative results and/or generic considerations. Additionally, I did not find the scientific purpose of the research and this article in the abstract. Significant improvements are expected in this section of the manuscript.

2) line 28, 32 and others - The Authors misspelled references in the main text of the manuscript. Currently it is [1][3] and it should be [13]. Please check and correct the entire text.

3) line 85 to 90 - The authors wrote: "This method reconstructs longitudinal wave velocity models from the measured ultrasonic data by extracting features from the component. In addition, many other works are performed using NDT methods such as ultrasonic guided waves, thermography, non‐linear ultrasound, laser ultrasound, eddy current to investigate the integrity of adhesive bonding [4], [8], [10], [16]–[27]."  The authors refer in one sentence to items in references 16 to 27. If these are important publications, e.g. affecting the authors' experimental concept, they should be described in detail (each separately in terms of research results, methodology or conclusions). The current provision of 12 references in one sentence is unacceptable and requires significant modifications. I also suggest that the authors verify other publications in the field of testing of adhesive joints using the ultrasonic method: e.g. http://dx.doi.org/10.3390/app122211782 and other publications published by this author regarding the ultrasonic assessment of adhesive joints.

4) At the end of paragraph I (Introduction), the novelty of the proposed solution and the scientific goal should be defined in detail. The article should also include a justification for the knowledge gap it fills. In its current form, the authors did not included the main scientific goal of presented research. There is some research on ultrasonic evaluation of adhesive joints so the authors must clearly indicate what they have done beyond what is already published in the literature.

5) Please justify the choice of the first 7 parameters presented in table 3 on page 7 of the manuscript. Why didn't the Authors use the reflection coefficient |r| parameter from different areas of the adhesive joint? This parameter characterizes the adhesion, in this case, e.g. between aluminum and adhesive.

6) Figure 7 should be deleted because the defect view has already been shown in Figure 1.

7) Please indicate the number of measurements from which the errors included in table 6 were determined. Why are the errors in Figure 9 and not the values of a given parameter with the designated error bar?

8) Authors present their results but without any discussion supported by the literature. When the results are not discussed and conveniently supported by the open literature, questionable conclusions are obtained. Currently, the article looks more like a report from simulation and experimental tests than a scientific article.

9) Research articles should present the directions of further research. I suggest adding one paragraph in the 4. Conclusion chapter.

10) Please read the instructions, how to describe the references at the end of the article in the Authors guide and change it. Currently, the references at the end of the text are not in line with the journal requirements. 

Taking into account the above comments, I believe the article needs major revision. I also believe that the topic presented in the article is relevant.  I hope these suggestions can help to improve the quality of this paper. I encourage the Authors to improve the manuscript according to the above remarks.

I wish you all the best.

Reviewer

Comments on the Quality of English Language

Dear Authors,

The article uses the personal form in several places, e.g. line 80: "He proposed...". This is not correct in high-quality articles. It suggests modifying this part of the article. Please check the entire article in terms of personal form.

Kind regards

Reviewer

Reviewer 2 Report

Comments and Suggestions for Authors

Citing information should not be packed, like in section 1 , line 90, 92for reference citations [16]-[27], [28]-[34] etc. It is not reasonable that all citations make identical statements. Build shorter groups of citations, if any. Only those papers that materially support or extend discussions of your work should be cited.

A numbers of abbreviation are used in the paper without using proper terminology to define/describe the physical terms, like LDA, SVM in line 70.

Several reference citations do not seem to match the title or work in the paper. Again, note that excessive and inappropriate self-citation or coordinated efforts among several authors to collectively self-cite is strongly discouraged. Please take this issue seriously.

In figure 3(a), some pulses in the first pulse response diagram do not correspond to the schematic diagram below. Please explain the reason in the article or modify the diagram.

An incorrect word ‘thev’ appeared in line 185.

Please unify the formula font in table 3, like ’N-’ in 9 line of table.

For the figure 6, the clarity of image is not good, and there are unnecessary color blocks and layers in the image.

In figure 8, what does the red dashed line in the two slice images mean? Please explain/emphasize in the article.

In this article, the ultrasonic features are from the C-scan results of No 3 sample, because the results of No 1 and No 4 sample are influenced by something, but the other two is not mentioned. It is not very reliable to analysis the features in one sample.

I recommend the paper for publication after the application of the above comments.

Round 2

Reviewer 1 Report

Comments and Suggestions for Authors

Dear Authors,

the manuscript entitled "Ultrasonic Features for Evaluation of Adhesive Joints: A Comparative Study of Interface Defects" by Damira Smagulova and co-authors has been significantly improved. Thank you for taking my suggestions into consideration and making modifications to the manuscript.

In my opinion, taking into account the Author's answers and corrections, I recommend the acceptance of the manuscript for publication.

Reviewer

Reviewer 2 Report

Comments and Suggestions for Authors

Dear Author:

The quality of the revised article has greatly improved.

But the clarity of figure.7 is not good, please modify it.

I recommend the paper for publication in this version.